# A Spatial-Temporal Attention-Based Method and a New Dataset for Remote Sensing Image Change Detection

**Hao Chen** [1,2,3] **and Zhenwei Shi** [1,2,3,*]

1   Image Processing Center, School of Astronautics, Beihang University, Beijing 100191, China; justchenhao@buaa.edu.cn
2   Beijing Key Laboratory of Digital Media, Beihang University, Beijing 100191, China
3   State Key Laboratory of Virtual Reality Technology and Systems, School of Astronautics, Beihang University, Beijing 100191, China
*   Correspondence: shizhenwei@buaa.edu.cn

**Abstract:** Remote sensing image change detection (CD) is done to identify desired significant changes between bitemporal images. Given two co-registered images taken at different times, the illumination variations and misregistration errors overwhelm the real object changes. Exploring the relationships among different spatial–temporal pixels may improve the performances of CD methods. In our work, we propose a novel Siamese-based spatial–temporal attention neural network. In contrast to previous methods that separately encode the bitemporal images without referring to any useful spatial–temporal dependency, we design a CD self-attention mechanism to model the spatial–temporal relationships. We integrate a new CD self-attention module in the procedure of feature extraction. Our self-attention module calculates the attention weights between any two pixels at different times and positions and uses them to generate more discriminative features. Considering that the object may have different scales, we partition the image into multi-scale subregions and introduce the self-attention in each subregion. In this way, we could capture spatial–temporal dependencies at various scales, thereby generating better representations to accommodate objects of various sizes. We also introduce a CD dataset LEVIR-CD, which is two orders of magnitude larger than other public datasets of this field. LEVIR-CD consists of a large set of bitemporal Google Earth images, with 637 image pairs (1024 × 1024) and over 31 k independently labeled change instances. Our proposed attention module improves the F1-score of our baseline model from 83.9 to 87.3 with acceptable computational overhead. Experimental results on a public remote sensing image CD dataset show our method outperforms several other state-of-the-art methods.

**Keywords:** image change detection; attention mechanism; multi-scale; spatial–temporal dependency; image change detection dataset; fully convolutional networks (FCN)

## 1. Introduction

Remote sensing change detection (CD) is the process of identifying "significant differences" between multi-temporal remote sensing images [1] (the significant difference usually depends on a specific application), which has many applications, such as urbanization monitoring [2,3], land use change detection [4], disaster assessment [5,6] and environmental monitoring [7]. Automated CD technology has facilitated the development of remote sensing applications and has been drawing extensive attention in recent years [8].

During the last few decades, many CD methods have been proposed. Most of these methods have two steps: unit analysis and change identification [8]. Unit analysis aims to build informative

features from raw data for the unit. The image pixel and image object are two main categories of the analysis unit [8]. Different forms of analysis units share similar feature extraction techniques. Spectral features [9–12] and spatial features [13,14] have been widely studied in CD literature. Change identification uses handcrafted or learned rules to compare the representation of analysis units to determine the change category. A simple method is to calculate the feature difference map and separate changing areas by thresholding [1]. Change vector analysis (CVA) [15] combines the magnitude and direction of the change vector for analyzing change type. Classifiers such as support vector machines (SVMs) [16,17] and decision trees (DTs) [13], and graphical models, such as Markov random field models [18,19] and conditional random field models [20], have also been applied in CD.

In addition to 2D spectral image data, height information has also been used for CD. For example, due to its invariance to illumination and perspective changes, the 3D geometric information could improve the building CD accuracy [21]. 3D information can also help to determine changes in height and volume, and has a wide range of applications, such as 3D deformation analysis in landslides, 3D structure and construction monitoring [22]. 3D data could be either captured by a light detection and ranging (LiDAR) scanner [21] or derived from geo-referenced images by dense image matching (DIM) techniques [23]. However, LiDAR data are usually expensive for large-scale image change detection. Image-derived 3D data is relatively easy to obtain from satellite stereo imagery, but it is relatively low quality. The reliability of the image-derived data is strongly dependent on the DIM techniques, which is still a major obstacle to accurate detection. In this work, we mainly focused on 2D spectral image data.

Most of the early attempts of remote sensing image CD are designed with the help of handcrafted features and supervised classification algorithms. The booming deep learning techniques, especially deep convolutional neural networks (CNN), which learn representations of data with multiple levels of abstraction [24], have been extensively applied both in computer vision [24] and remote sensing [25]. Nowadays, many deep-learning-based CD algorithms [26–33] have been proposed and demonstrate better performances than traditional methods. These methods can be roughly divided into two categories: metric-based methods [26–28,33] and classification-based methods [29–32,34].

Metric-based methods determine the change by comparing the parameterized distance of bitemporal data. These methods need to learn a parameterized embedding space, where the embedding vectors of similar (no change) samples are encouraged to be closer, while dissimilar (change) ones are pushed apart from each other. The embedding space can be learned by deep Siamese fully convolutional networks (FCN) [27,28], which contains two identical networks sharing the same weight, each independently generating the feature maps for each temporal image. A metric (e.g., L1 distance) between features of each pair of points is used to indicate whether changes have occurred. For the training process, to constrain the data representations, different loss functions have been explored in CD, such as contrastive loss [26,27,33] and triplet loss [28]. The method using the triplet loss achieves better results than the method using the contrast loss because the triplet loss exploits more spatial relationships among pixels. However, existing metric-based methods have not utilized temporal dependency between bitemporal images.

Classification-based methods identify the change category by classifying the extracted bitemporal data features. A general approach is to assign a change score to each position of the image, where the position of change has a higher score than that of no change. CNN has been widely used for extracting feature representations for images [29–31,34]. Liu et al. [34] developed two approaches based on FCN to detect the slums change, including post-classification and multi-date image classification. The first approach used an FCN to separately classify the land use of each temporal image and then determined the change type by the change trajectory. The second approach concatenated the bitemporal images and then used an FCN to obtain the change category. It is important to extract more discriminative features for the bitemporal images. Liu et al. [30] utilized spatial and channel attention to obtain more discriminative features when processing each temporal image. These separately extracted features of each temporal image were concatenated to identify changes. Temporal dependency between

bitemporal images has not been exploited. A recurrent neural network (RNN) is good at handling sequential relationships and one has been applied in CD to model temporal dependency [29,31,32]. Lyu et al. [32] employed RNN to learn temporal features from bitemporal sequential data, but spatial information has not been utilized. To exploit spatial–temporal information, several methods [29,31] combined CNN and RNN to jointly learn the spatial–temporal features from bitemporal images. These methods inputted very small image patches (e.g., size of 9 × 9) into CNN to obtain 1D feature representations for the center point of each patch, then used RNN to learn the relationships between the two points at different dates. The spatial–temporal information utilized by these methods is very limited.

Given two co-registered images taken at different times, the illumination variations and misregistration errors caused by the change of sunlight angle overwhelm the real object changes, challenging the CD algorithms. From Figure 1a, we can observe that the image contrast and brightness, as well as the buildings' spatial textures, are different in the two images. There are different shadows along with the buildings in the bitemporal images caused by solar position change. Figure 1b shows that misregistration errors are significant at the edge of corresponding buildings in the two co-registered images. If not treated carefully, they can cause false detections at the boundaries of unchanged buildings.

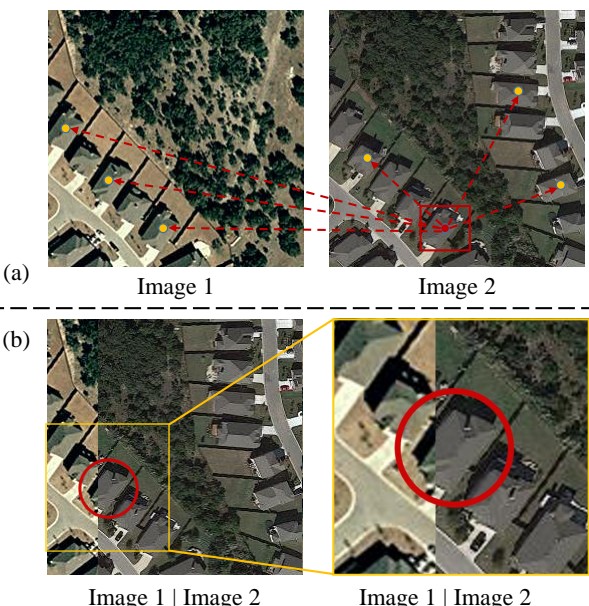

**Figure 1.** (**a**) Illustrations of spatial–temporal attention. The red lines denote selected spatial–temporal attention. (**b**) Illustrations of misregistration errors. (Zoom in for a better view.)

In this paper, we design a CD self-attention mechanism, which captures rich spatial–temporal relationships to obtain illumination-invariant and misregistration-robust features. The motivations of our method derive from the following two aspects:

(1) Since the CD data are composed of spectral vectors both in the temporal and spatial dimensions, exploring the relationships among different spatial–temporal positions may improve the performances of CD methods. For example, by exploiting the relationships among objects of the same kind at different times and locations, networks may produce similar feature representations for these objects, despite their illumination differences. The impact of misregistration errors may be reduced by utilizing the global relationship between the objects at different times. Moreover, modeling the spatial–temporal relationships among neighbor pixels is proven useful in CD [29]. The self-attention mechanism is effective in modeling long-range spatial–temporal dependencies [35]. Inspired by this recognition, we integrate a new CD self-attention module in the procedure of feature extraction, so as to generate a more powerful network.

(2) Since the object of change may have different scales, extracting features from a suitable scope may better represent the object of a certain scale. We could obtain multi-scale features by combining features extracted from regions of different sizes. Driven by this motivation, we divide the image space equally into subregions of a certain scale, and introduce the self-attention mechanism in each subregion to exploit the spatial–temporal relationship for the object at this scale. By partitioning the image into subregions of multiple scales, we can obtain feature representations at multiple scales to better adapt to the scale changes of the object. We call this architecture pyramid attention module because the self-attention module is integrated into the pyramid structure of multi-scale subregions. In this way, we could capture spatial–temporal dependencies at various scales, thereby generating better representations to accommodate objects of various sizes.

Based on the above motivations, our solution comes as no surprise. We propose a spatial–temporal attention neural network (STANet) for CD, which belongs to the metric-based method. The Siamese FCN is employed to extract the bitemporal image feature maps. Our self-attention module updates these feature maps by exploiting spatial–temporal dependencies among individual pixels at different positions and times. When computing the response of a position of the embedding space, the position pays attention to other important positions in space-time by utilizing the spatial–temporal relationship. Here, the embedding space has the dimensions of height, width and time, which means a position in the embedding space can be described as (h,w,t). For simplicity, we denote the embedding space as space-time. As illustrated in Figure 1a, the response of the pixel (belongs to building) in the red bounding box pays more attention to the pixels of the same category in the whole space-time, which indicates that pixels of the same category have strong spatial–temporal correlations; such correlations could be exploited to generate more discriminative features.

We design two kinds of self-attention modules: a basic spatial–temporal attention module (BAM) and a pyramid spatial–temporal attention module (PAM). BAM learns to capture the spatial–temporal dependency (attention weight) between any two positions and compute each position's response by the weighted sum of the features at all the positions in the space-time. PAM embeds BAM into a pyramid structure to generate multi-scale attention representations. See Section 2.1.3 for more details of BAM and PAM.

Our method is different from previous deep learning-based remote sensing CD algorithms. Previous metric-based methods [27,28,33] separately process bitemporal image series from different times without referring to any useful temporal dependency. Previous classification-based methods [29–31] also do not fully exploit the spatial–temporal dependency. Those RNN-based methods [29,31] introduce RNN to fuse 1D feature vectors from different times. The feature vectors are obtained from very small image patches through CNN. On the one hand, due to the small image patch, the extracted spatial features are very limited. On the other hand, the spatial–temporal correlations among individual pixels at different positions and times have not been utilized. In this paper, we design a CD self-attention mechanism to exploit the explicit relationships between pixels in space-time. We could visualize the attention map to see what dependencies are learned (see Section 4). Unlike previous methods, our attention module can capture long-range, rich spatial–temporal relationships. Moreover, we integrate a self-attention module into a pyramid structure to capture spatial–temporal dependencies of various scales.

**Contributions.** The contributions of our work can be summarized as follows:

(1)　We propose a new framework; namely, a spatial–temporal attention neural network (STANet) for remote sensing image CD. Previous methods independently encode bitemporal images, while in our framework, we design a CD self-attention mechanism, which fully exploits the spatial–temporal relationship to obtain illumination-invariant and misregistration-robust features.

(2)　We propose two attention modules: a basic spatial–temporal attention module (BAM) and a pyramid spatial–temporal attention module (PAM). The BAM exploits the global spatial–temporal relationship to obtain better discriminative features. Furthermore, the PAM aggregates multi-scale

attention representation to obtain finer details of objects. Such modules can be easily integrated with existing deep Siamese FCN for CD.

(3) Extensive experiments have confirmed the validity of our proposed attention modules. Our attention modules well mitigate the misdetections caused by misregistration in bitemporal images and are robust to color and scale variations. We also visualize the attention map for a better understanding of the self-attention mechanism.

(4) We introduce a new dataset LEVIR-CD (LEVIR building Change Detection dataset), which is two orders of magnitude larger than existing datasets. Note that LEVIR is the name of the authors' laboratory: the Learning, Vision and Remote Sensing Laboratory. Due to the lack of a public, large-scale CD dataset, the new dataset should push forward the remote sensing image CD research. We will make LEVIR-CD open access at https://justchenhao.github.io/LEVIR/. Our code will also be open source.

## 2. Materials and Methods

In this section, we first present a detailed description of our proposed method, then introduce a new remote sensing image CD dataset. Finally, the experimental implementation details are given.

### 2.1. STANet: Spatial–Temporal Attention Neural Network

In this subsection, the overall pipeline of our method is given. Then, a detailed description of the proposed spatial–temporal attention neural network (STANet) is provided.

#### 2.1.1. Overview

Given the bitemporal remote sensing images $I^{(1)}, I^{(2)}$ of size $H_0 \times W_0$, the goal of CD is generating a label map $M$, the same size of the input images, where each spatial location is assigned to one change type. In this work, we focus on binary CD, which means the label is either 1 (change) or 0 (no change).

The pipeline of our method is shown in Figure 2a. The spatial–temporal attention neural network has three components: a feature extractor (see Section 2.1.2), an attention module (Section 2.1.3) and a metric module (Section 2.1.4). First, the two images are sequentially fed into the feature extractor (an FCN, e.g., the ResNet [36] without fully connected layers) to obtain two feature maps $X^{(1)}, X^{(2)} \in \mathbb{R}^{C \times H \times W}$, where $H \times W$ is the size of the feature map and $C$ is the channel dimension of each feature vector. These feature maps are then updated to two attention feature maps $Z^{(1)}, Z^{(2)}$ by the attention module. After resizing the updated feature maps to the size of the input images, the metric module calculates the distance between each pixel pair in the two feature maps and generates a distance map $D$. In the training phase, our model is optimized by minimizing the loss calculated by the distance map and the label map, such that the distance value of the change point is large and the distance value of the no-change point is small. While in the testing phase, the predicted label map $P$ can be calculated by simple thresholding on the distance map.

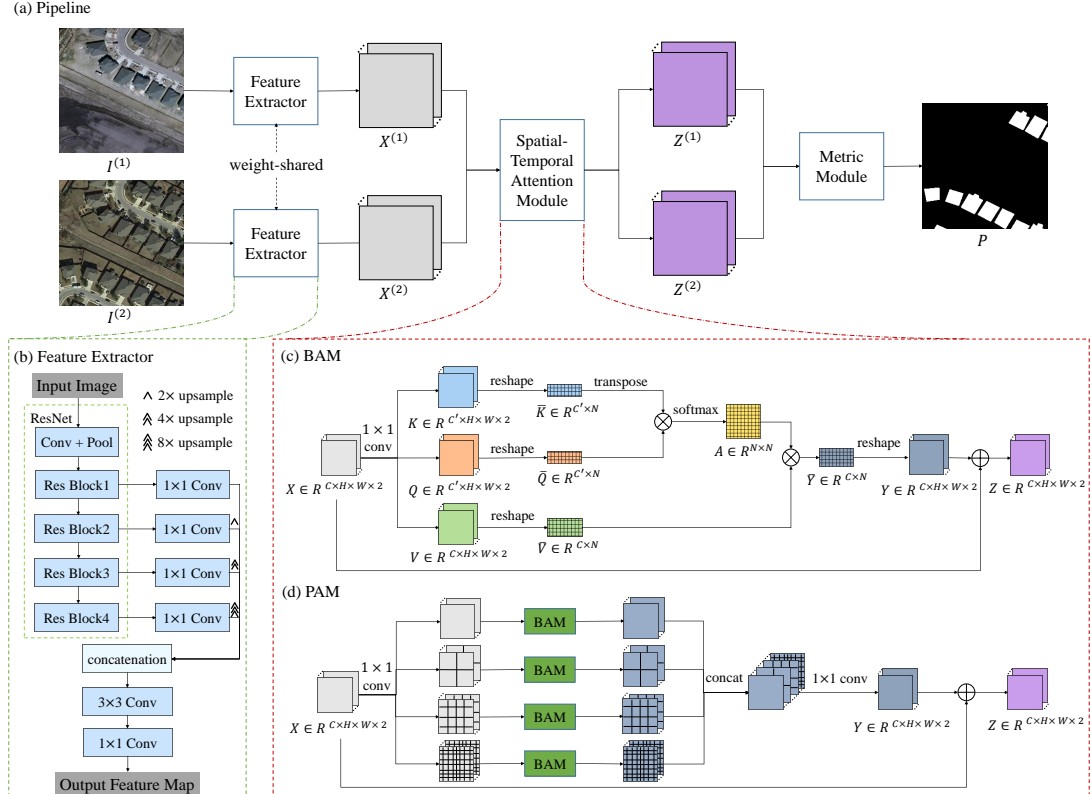

**Figure 2.** (**a**) The Pipeline of STANet. Note that we have designed two kinds of self-attention modules. (**b**) Feature extractor. (**c**) Basic spatial–temporal attention module (BAM). (**d**) Pyramid spatial–temporal attention module (PAM).

### 2.1.2. Feature Extractor

During the last few years, many effective convolutional neural networks (CNN) [36–38] have been proposed for learning better features, which greatly surpass traditional handcrafted features in various visual tasks. In light of the good performance in computer vision, deep-CNN-based methods have been widely applied in remote sensing tasks [25], such as land-use classification [39], semantic segmentation [40,41], image super-resolution [42], object detection [43,44] and change detection [5]. FCN [45] is a kind of CNN without fully connected layers, which is widely used for dense classification tasks. Remote sensing image CD requires pixel-wise prediction and benefits from the dense features by FCN based methods [46,47]. Our work borrows ResNet [36] for constructing the feature extractor.

As illustrated in Figure 2b, we have designed an FCN-like feature extractor. Our feature extractor is based on ResNet-18 [36]. Because the original ResNet is designed for the image classification task, it contains a global pooling layer and a fully connected layer at the end for mapping the image features to a 1000-dimensional vector (the number of categories in ImageNet). Our CD task is a dense classification task, which needs to obtain a change mask the same size as the input image. Therefore, we omit the global pooling layer and the fully connected layer of the original ResNet. The remaining part has five stages; each has a stride of 2. The high-level features in CNN are accurate in semantics but coarse in location, while the low-level features contain fine details but lack semantic information. Therefore, we fuse the high-level semantic information and low-level spatial information to generate finer representations. We take the output feature map of the last stage and feed it into a convolution layer ($C_1$, $1 \times 1/1$) to transform its dimensions to $C_1$. Note that the configuration of a convolution layer is "number of the filters, size/stride" and the batch normalization (BN) and ReLU layers are omitted for simplicity. Similarly, the output feature maps of the 2nd, 3rd and 4th stages are fed into three different convolution layers respectively; each channel dimension is converted to $C_1$. Then, we resize the transformed feature maps of the last three stages to the 1/4 size of the input images. In this

way, we obtain 4 sets of feature maps from different stages of the networks. These four feature maps are concatenated in the channel dimension (result in $4 \times C_1$) and fed into two different convolution layers ($C_2, 3 \times 3/1$; $C_3, 1 \times 1/1$) to generate the final feature map. These two convolution layers can generate more discriminative and compact representations by exploiting local spatial information and reducing feature channel dimensionality. In the implementation, we set $C_1$ to 96, $C_2$ to 256, and $C_3$ to 64 for a trade-off between efficiency and accuracy.

### 2.1.3. Spatial–Temporal Attention Module

Originating from the human visual system [48], attention mechanism models the dependencies between input and output sequences and has been applied in various tasks, such as neural machine translation [49], image captioning [50] and scene parsing [51]. Self-attention [52] is an attention mechanism relating different positions of a single sequence to calculate the representation of each position of the sequence. The non-local neural networks [35] extended the self-attention mechanism in many computer vision tasks, such as video classification, object detection and pose estimation. The self-attention mechanism is effective in modeling long-range spatial–temporal dependencies [35]. Motivated by this recognition, we design a CD self-attention mechanism, which captures the rich global spatial–temporal relationships among individual pixels in the whole space-time to obtain more discriminative features. Concretely, we propose two kinds of spatial–temporal attention module; namely, the basic spatial–temporal attention module (BAM) and the pyramid spatial–temporal attention module (PAM). Their detailed descriptions are as follows:

**Basic spatial–temporal attention module:**

To illustrate the basic idea of the self-attention mechanism, we introduce three terms: query, key and value [52]. Suppose we have a database with many key-value pairs. For a new query, we need to find the element that matches it best in the database. We can achieve that by calculating the similarity between the query and all the keys in the database. The self-attention mechanism is based on this idea to calculate the correlations between different elements. In particular, in the self-attention mechanism, the query and key are obtained from the same source.

In this paper, let a query (or key, value) denote a vector of a certain position in the query (or key, value) tensor, where the query, key and value tensors are separately obtained from the input feature tensor through three different convolutional layers (The feature tensor is the concatenation of the bitemporal image feature maps in the temporal dimension). The core of the spatial–temporal attention module is to learn an attention function, which maps a query vector and a set of key-value vector pairs to an output vector. The output vector is computed by the weighted sum of the value vectors, where the weight assigned to each value vector is calculated by an affinity function of the query and the corresponding key. Through the self-attention module, we can obtain the output feature tensor, where each position can attend to all positions in the input feature tensor. The intuition of introducing the self-attention mechanism to image CD is that fully exploiting the spatial–temporal dependencies between pixels may help obtain illumination-invariant and misregistration-robust features.

Figure 2c illustrates the details of the BAM. We stack the bitemporal feature maps $X^{(1)}, X^{(2)}$ into a feature tensor $X \in \mathbb{R}^{C \times H \times W \times 2}$; then feed it into the BAM to produce the updated feature tensor $Z \in \mathbb{R}^{C \times H \times W \times 2}$; and finally split it into two feature maps $Z^{(1)}, Z^{(2)}$. Here, we employ a residual function to derive $Z$ from the input $X$:

$$Z = \mathcal{F}(X) + X \tag{1}$$

where $Y = \mathcal{F}(X)$ is a residual mapping of $X$ to be learned.

The core of calculating $Y$ is to generate a set of key vectors (keys), value vectors (values) and query vectors (queries) from the input tensor, and learn the weighted sum of the values to generate each output vector, where the weight assigned to each value depends on the similarity of the query and the corresponding key. Now, we give a detailed description of the process of calculating the residual mapping $Y$. First, we calculate the keys, values and queries from the input. The input feature

tensor $X$ is firstly transformed into two feature tensors $Q, K \in \mathbb{R}^{C' \times H \times W \times 2}$. Q and K are respectively obtained by two different convolution layers $(C', 1 \times 1/1)$. We reshape them into a key matrix $\bar{K}$ and a query matrix $\bar{Q} \in \mathbb{R}^{C' \times N}$, where $N = H \times W \times 2$ is the number of the input feature vectors. The key matrix and the query matrix are used to calculate the attention later. Similarly, we feed $X$ into another convolution layer $(C, 1 \times 1/1)$, to generate a new feature tensor $V \in \mathbb{R}^{C \times H \times W \times 2}$. We reshape it into a value matrix $\bar{V} \in \mathbb{R}^{C \times N}$. $C'$ is the feature dimension of the keys and the queries. In our implementation, $C'$ is assigned to $\frac{C}{8}$ for reducing the feature dimension.

Secondly, we define the spatial–temporal attention map $A \in \mathbb{R}^{N \times N}$ as the similarity matrix. The element $A[i, j]$ in similarity matrix is the similarity between the $i$th key and the $j$th query. We perform a matrix multiplication between the transpose of the key matrix $\bar{K}^T$ and the query matrix $\bar{Q}$, divide each element by $\sqrt{C'}$ and apply a softmax function to each column to generate the attention map $A$. $A$ is defined as follows:

$$A = softmax(\frac{\bar{K}^T \bar{Q}}{\sqrt{C'}}) \tag{2}$$

Note that the matrix multiplication result is scaled by $\sqrt{C'}$ for normalizing its expected value from being affected by large values of $C'$ [52].

Finally, the output matrix $\bar{Y} \in \mathbb{R}^{C \times N}$ is computed by the matrix multiplication of the value matrix $\bar{V}$ and the similarity matrix $A$:

$$\bar{Y} = \bar{V} A \tag{3}$$

$\bar{Y}$ is then reshaped into $Y$.

**Pyramid spatial–temporal attention module:**

Context plays an important role in many visual tasks, such as video surveillance [53], semantic segmentation [46,54] and object detection [55]. PSPNet [54] exploits the global spatial context information by different-region-based context aggregation. As for remote sensing image CD, the spatial and temporal context was discussed in [29,56,57]. Inspired by the pyramid structure of PSPNet [54], we propose a PAM to enhance the ability to identify fine details by aggregating the multi-scale spatial–temporal attention context. PAM generates multi-scale attention features by combining the spatial–temporal attention context of different scales. The PAM has four branches; each branch partitions the feature tensor equally into several subregions of a certain scale. In each branch, the PAM applies BAM to pixels in each subregion to obtain the local attention representation at this scale. Then, the multi-scale attention representation is generated by aggregating the output tensor of the four branches. We call this architecture a pyramid attention module, because every pixel in the image space is involved in self-attention mechanisms in subregions of different scales. It can be imagined that these subregions are arranged from small to large, just like the structure of a pyramid.

Figure 2d gives an illustration of the PAM. Given the bitemporal feature maps $X^{(1)}, X^{(2)} \in \mathbb{R}^{C \times H \times W}$, we stack the two feature maps into a feature tensor $X \in \mathbb{R}^{C \times H \times W \times 2}$. Then we have four parallel branches; each branch partitions the feature tensor equally into $s \times s$ subregions, where $s \in S$. $S = \{1, 2, 4, 8\}$ defines four pyramid scales. In the branch of scale $s$, each region is defined as $R_{s,i,j} \in \mathbb{R}^{C \times \frac{H}{s} \times \frac{W}{s} \times 2}, 1 \leq i, j \leq s$. We employ four BAMs to the four branches separately. Within each pyramid branch, we apply the BAM to all the subregions $R_{s,i,j}$ separately to generate the updated residual feature tensor $Y_s \in \mathbb{R}^{C \times H \times W \times 2}$. Then, we concatenate these feature tensors $Y_s, s \in S$, and feed them into a convolution layer $(C, 1 \times 1/1)$ to generate the final residual feature tensor $Y \in \mathbb{R}^{C \times H \times W \times 2}$. Finally, we add the residual tensor $Y$ and the original tensor $X$ to produce the updated tensor $Z \in \mathbb{R}^{C \times H \times W \times 2}$.

### 2.1.4. Metric Module

Deep metric learning involves training a network to learn a nonlinear transformation from input to the embedding space [58], where the embedding vectors of similar samples are encouraged to

be closer, while dissimilar ones are pushed apart from each other [59]. During the last few years, deep metric learning has been applied in many remote sensing applications [27,28,60,61]. Deep metric learning-based CD methods [27,28] achieved the leading performance. Here, we employed a contrastive loss to encourage a small distance ofor each no-change pixel pair and a large distance for each change in the embedding space.

Given the updated feature maps $Z^{(1)}, Z^{(2)}$, we firstly resize each feature map to be the same size as the input bitemporal images by bilinear interpolation. Then we calculate the euclidean distance between the resized feature maps pixel-wise to generate the distance map $D \in \mathbb{R}^{H_0 \times W_0}$, where $H_0, W_0$ are the height and width of the input images respectively. In the training phase, a contrastive loss is employed to learn the parameters of the network, in such a way that neighbors are pulled together and non-neighbors are pushed apart. We will give a detailed definition of the loss function in Section 2.1.5. While in the testing phase, the change map $P$ is obtained by a fixed threshold segmentation:

$$P_{i,j} = \begin{cases} 1 & D_{i,j} > \theta \\ 0 & \text{else.} \end{cases} \tag{4}$$

where the subscript $i, j$ ($1 \leq i \leq H_0, 1 \leq j \leq W_0$) denote the indexes of the height and width respectively. $\theta$ is a fixed threshold to separate the change areas. In our work, $\theta$ is assigned to 1, which is half of the margin defined in the loss function.

### 2.1.5. Loss Layer Design

CThe cass imbalance problem is common in most machine learning tasks where the class distribution is highly imbalanced [62]. For remote sensing image CD, the numbers of change and no-change samples vary greatly. In many cases, the change pixels only make up a small fraction of all pixels, which causes some bias in the network during the training phase. To reduce the impact of the class imbalance, we design a class-sensitive loss; namely, batch-balanced contrastive loss (BCL). It utilizes the batch-weight prior to modify the class weights of the original contrastive loss [63]. Given a batch bitemporal samples $(X^{*(1)}, X^{*(2)}, M^*), X^{*(1)}, X^{*(2)} \in \mathbb{R}^{B \times 3 \times H_0 \times W_0}, M^* \in \mathbb{R}^{B \times H_0 \times W_0}$, we can obtain a batch of distance maps $D^* \in \mathbb{R}^{B \times H_0 \times W_0}$ through the STANet, where $B$ is the batch size of the samples. $M^*$ is a batch of binary label maps, where 0 denotes no change and 1 represents a change. The BCL $L$ is defined as follows:

$$\begin{aligned} L(D^*, M^*) = &\frac{1}{2} \frac{1}{n_u} \sum_{b,i,j} (1 - M^*_{b,i,j}) D^*_{b,i,j} \\ &+ \frac{1}{2} \frac{1}{n_c} \sum_{b,i,j} M^*_{b,i,j} Max(0, m - D^*_{b,i,j}) \end{aligned} \tag{5}$$

where the subscript $b, i, j$ ($1 \leq b \leq B, 1 \leq i \leq H_0, 1 \leq j \leq W_0$) denote the indexes of the batch, height and width respectively. The change pixel pair whose parameterized distance is larger than the margin $m$ does not contribute to the loss function. In our work, $m$ is set to 2. $n_u, n_c$ are the numbers of the no change pixel pairs and the changed ones respectively. They can be calculated by the sum of the labels of the corresponding category:

$$n_u = \sum_{b,i,j} 1 - M^*_{b,i,j} \tag{6}$$

$$n_c = \sum_{b,i,j} M^*_{b,i,j} \tag{7}$$

### 2.2. LEVIR-CD: A New Remote Sensing Image Change Detection Dataset

Large and challenging datasets are very important for remote sensing applications. However, in remote sensing image CD, we notice the lack of a public, large-scale CD dataset, which discourages

the research of CD, especially for developing deep-learning-based algorithms. Therefore, through introducing the LEVIR-CD dataset, we want to fill this gap and provide a better benchmark for evaluating the CD algorithms.

We collected 637 very high-resolution (VHR, 0.5 m/pixel) Google Earth (GE) image patch pairs with a size of 1024 × 1024 pixels via Google Earth API. These bitemporal images are from 20 different regions that sit in several cities in Texas of the US, including Austin, Lakeway, Bee Cave, Buda, Kyle, Manor, Pflugervilletx, Dripping Springs, etc. Figure 3 illustrates the geospatial distribution of our new dataset and an enlarged image patch. Each region has a different size and contains a varied number of image patches. Table 1 lists the area and number of patches for each region. The capture-time of our image data varied from 2002 to 2018. Images in different regions may be taken at different times. We want to introduce variations due to seasonal changes and illumination changes into our new dataset, which could help develop effective methods that can mitigate the impact of irrelevant changes on real changes. The specific capture-time of each image in each region is listed in Table 1. These bitemporal images have a time span of 5 ∼ 14 years.

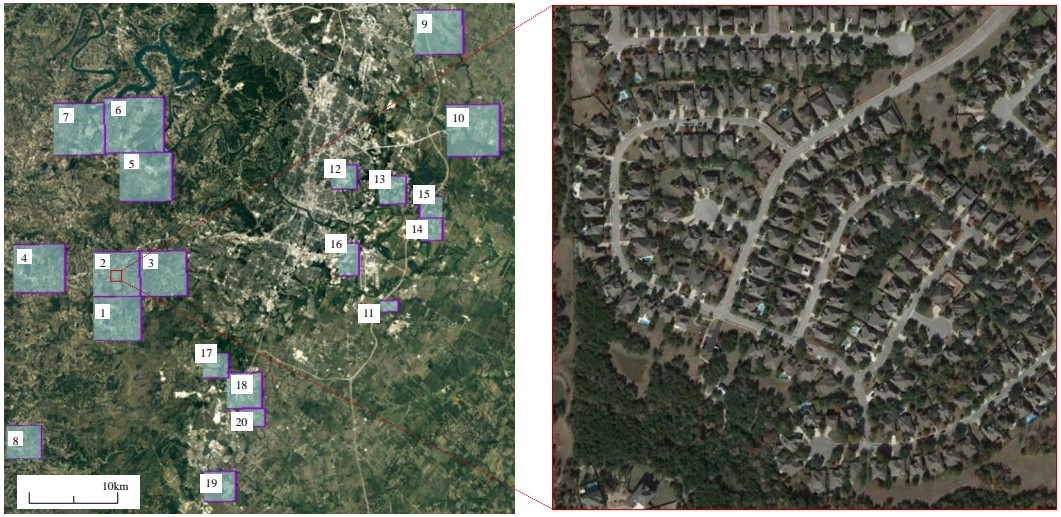

**Figure 3.** Left: geospatial distribution of our new dataset. Right: an enlarged image patch.

**Table 1.** Details of each region in our collected dataset.

| Region ID | Area (km$^2$) | Number of Patch Pairs | Time 1 (Year/Month) | Time 2 (Year/Month) |
|---|---|---|---|---|
| 1 | 6.8 | 26 | 2002/12 | 2013/11 |
| 2 | 5.5 | 21 | 2002/12 | 2013/11 |
| 3 | 5.5 | 21 | 2002/12 | 2013/11 |
| 4 | 5.0 | 19 | 2002/02 | 2017/11 |
| 5 | 8.1 | 31 | 2003/03 | 2017/01 |
| 6 | 10.7 | 41 | 2003/03 | 2017/01 |
| 7 | 6.0 | 23 | 2017/11 | 2018/06 |
| 8 | 11.5 | 44 | 2008/02 | 2018/01 |
| 9 | 26.4 | 94 | 2006/04 | 2017/01 |
| 10 | 10.2 | 39 | 2003/03 | 2017/01 |
| 11 | 2.1 | 8 | 2003/03 | 2015/07 |
| 12 | 7.9 | 30 | 2009/03 | 2017/02 |
| 13 | 7.9 | 30 | 2003/03 | 2017/01 |
| 14 | 5.2 | 20 | 2003/03 | 2012/08 |
| 15 | 5.2 | 20 | 2003/03 | 2013/11 |
| 16 | 6.3 | 24 | 2006/04 | 2016/02 |
| 17 | 7.9 | 30 | 2009/02 | 2017/01 |
| 18 | 14.7 | 56 | 2009/02 | 2017/01 |
| 19 | 11.0 | 42 | 2011/03 | 2017/01 |
| 20 | 4.7 | 18 | 2012/08 | 2017/01 |

Buildings are representative of man-made structures. Detecting the change of buildings is an important CD task with various applications, such as urbanization monitoring and illegal building identification. During the last few decades, our collected areas have seen significant land-use changes, especially urban constructions. The VHR remote sensing images provide an opportunity for us to analyze the subtle changes, such as the changes of building instances. Therefore, we focus on building-related changes, including the building growth (the change from soil/grass/hardened ground or building under construction to new build-up regions) and the building decline. Our new dataset covers various types of buildings, such as villa residences, tall apartments, small garages and large warehouses. Some selected samples from the dataset are shown in Figure 4, which displays several samples of building updating, building decline and no change. We can observe that there are a variety of buildings in our dataset.

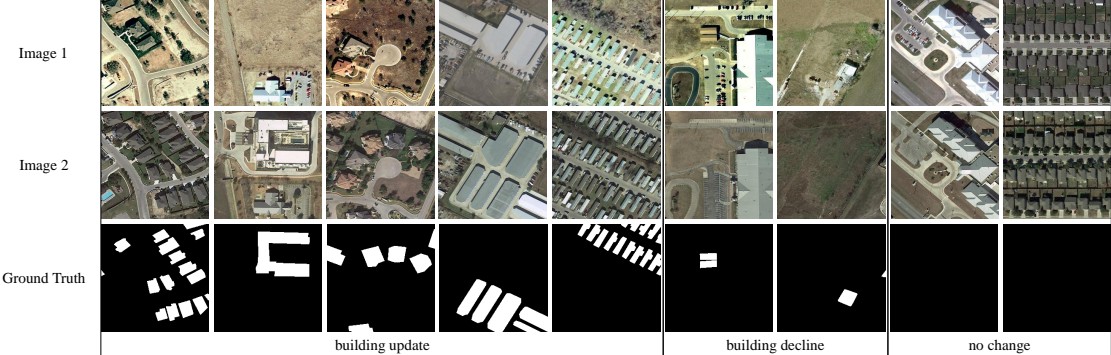

**Figure 4.** Selected cropped samples (256 × 256) from LEVIR-CD. Each column represents one sample, including the image pair (row 1 and 2), and the label (the last row, white denotes change, black means no change). Columns 1–5 show the building update; 6 and 7 show the building decline; and the last two columns display samples of no change.

The bitemporal images were annotated by remote sensing image interpretation experts who are employed by an AI data service company (Madacode: http://www.madacode.com/index-en.html). All annotators had rich experience in interpreting remote sensing images and a comprehensive understanding of the change detection task. They followed detailed specifications for annotating images to obtain consistent annotations. Moreover, each sample in our dataset was annotated by one annotator and then double-checked by another to produce high-quality annotations. Annotation if such a large-scale dataset is very time-consuming and laborious. It takes about 120 person-days to manually annotate the whole dataset.

The fully annotated LEVIR-CD contains a total of 31,333 individual change buildings. On average, there are about 50 change buildings in each image pair. It is worth noting that most changes are due to the construction of new buildings. The average size of each change area is around 987 pixels. Table 2 provides a summary of our dataset.

**Table 2.** A summary of LEVIR-CD dataset.

| Type | Item | Value |
|---|---|---|
| Image Info. | # Total Image Pairs | 637 |
| | Image Size | 1024 × 1024 |
| | Image Resolution | 0.5 m/pixel |
| | Time Span | 5~14 years |
| | Modality | RGB image |
| Change Info. | # Total Change Instances | 31,333 |
| | # Total Change Pixels | 30,913,975 |
| | Average Change Size | 987 pixels |

Google Earth images are free to the public and have been used to promote many remote sensing studies [64,65]. However, the use of the images from Google Earth must respect the Google Earth terms of use (https://www.google.com/permissions/geoguidelines/). All images and annotations in LEVIR-CD can only be used for academic purposes, and are prohibited for any commercial use. There are two reasons for utilizing GE images: (1) GE provides free VHR historical images for many locations. We could choose one appropriate location and two appropriate time-points during which many significant changes have occurred at the location. In this way, we could collect large-scale bitemporal GE images. (2) We could collect diversified Google data to construct challenging CD datasets, which contain many variations in sensor characteristics, atmospheric conditions, seasonal conditions and illumination conditions. It would help develop CD algorithms that can be invariant to irrelevant changes but sensitive to real changes. Our dataset also has limitations. For example, our images have relatively poor spectral information (i.e., red, green and blue) compared to some other multispectral data (e.g., Landsat data). However, our VHR images provide fine texture and geometry information, which to some extent compensates for the limitation of poor spectral characteristics.

During the last few decades, some efforts have been made toward developing public datasets for remote sensing image CD. Here, let us provide a brief overview of three CD datasets:

**SZTAKI AirChange Benchmark Set (SZTAKI)** [66] is a binary CD dataset which contains 13 optical aerial image pairs; each is $952 \times 640$ pixels and resolution is about 1.5 m/pixel. The dataset is split into three sets by regions; namely, Szada, Tiszadob and Archive; they contain 7, 5 and 1 image pairs, respectively. The dataset considers the following changes: new built-up regions, building operations, planting forests, fresh plough-lands and groundwork before building completion.

**The Onera Satellite Change Detection dataset (OSCD)** [67] is designed for binary CD with the collection of 24 multi-spectral satellite image pairs. The size of each image is approximately $600 \times 600$ at 10 m resolution. The dataset focuses on the change of urban areas (e.g., urban growth and urban decline) and ignores natural changes.

**The Aerial Imagery Change Detection dataset (AICD)** [68] is a synthetic binary CD dataset with 100 simulated scenes, each captured from five viewpoints giving a total of 500 images. Each image was added with one kind of artificial change target (e.g., buildings, trees or relief) to generate the image pair. Therefore, there is one change instance in each image pair.

The comparison of our dataset and other remote sensing image CD datasets is shown in Table 3. SZTAKI is the most widely used public remote sensing image CD dataset and has helped impelled many recent advances [27,28,69]. OSCD, introduced last year, has also driven a few studies [67,69]. AICD has also helped in developing CD algorithms [70]. However, these existing datasets have many shortcomings. Firstly, all these datasets do not have enough data for supporting most deep-learning-based CD algorithms, which are prone to suffer overfitting problems when the data quantity is much to scare for the number of the model parameters. Secondly, these CD datasets have low image resolution, which blurs the contour of the change targets and brings ambiguity to annotated images. We count the numbers of change instances and change pixels of these datasets, which shows that our dataset is 1 ~ 2 orders of magnitude larger than existing datasets. As illustrated in Figure 5, we created a histogram the sizes of all the change instances of LEVIR-CD and SZTAKI. We can observe that the range of change instances size of our dataset is wider than that of SZTAKI, and LEVIR-CD contains far more change instances than SZTAKI.

**Table 3.** A comparison of LEVIR-CD and other remote sensing image change detection datasets.

| Dataset | # Pairs | # Size | Is Real? | # Change Instances | # Change Pixels |
|---|---|---|---|---|---|
| SZTAKI [66] | 12 | $952 \times 640$ | ✓ | 382 | 412,252 |
| OSCD [67] | 24 | $600 \times 600$ | ✓ | 1048 | 148,069 [1] |
| AICD [68] | 500 | $600 \times 800$ | × | 500 | 203,355 |
| LEVIR-CD | 637 | $1024 \times 1024$ | ✓ | 31,333 | 30,913,975 |

[1] As the authors of the OSCD dataset have not provided the ground truth of the test samples (10 samples), the numbers of change instances and change pixels are the statistical results on the training set (14 samples).

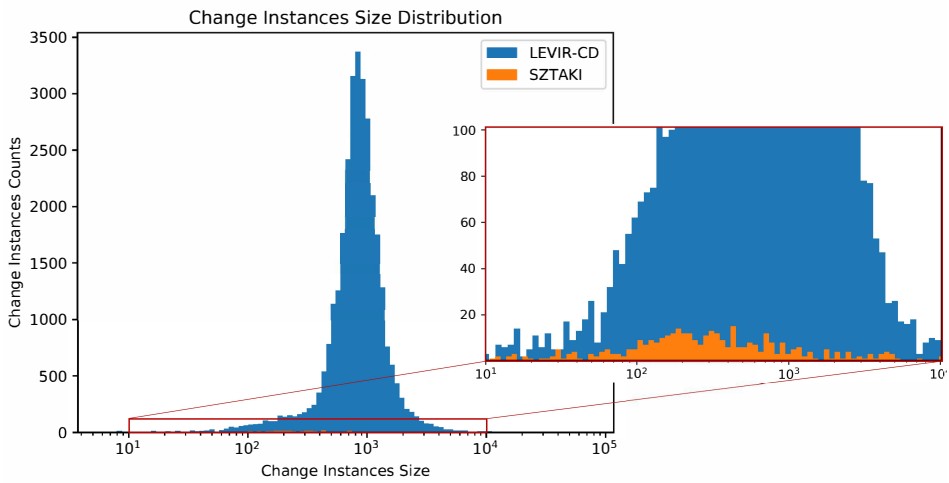

**Figure 5.** Distribution of change instance sizes (pixels) of LEVIR-CD and SZTAKI.

## 2.3. Implementation Details

**Metrics.** We regard the precision (Pr), recall (Re) and F1-score (F1) as evaluation metrics. Let $n_{ij}$ be the number of pixels of class $i$ predicted as class $j$, where there are $n_c$ classes. We computed:

- precision of class $i$ ($Pr_i$): $n_{ii} / \sum_j n_{ji}$.
- recall of class $i$ ($Re_i$): $n_{ii} / \sum_j n_{ij}$.
- F1-score of class $i$ ($F1_i$): $2Pr_i Re_i / (Pr_i + Re_i)$.

Specifically, we adopt precision, recall and F1-score related to the change category as evaluation metrics.

**LEVIR-CD dataset.** We randomly split the dataset into three parts—70% samples for training, 10% for validation and 20% for testing. Due to the memory limitation of GPU, we crop each sample to 16 small patches of size of $256 \times 256$.

**SZTAKI dataset.** We use the same training-testing split criterion as other comparison methods. The test set consists of patches of the size of $784 \times 448$ cropped from the top-left corner in each sample. The remaining part of each sample is clipped overlappingly into small patches of the size of $113 \times 113$ as training data.

**Training details.** We implement our methods based on Pytorch [71]. Our models are fine-tuned on the ImageNet-pre-trained ResNet-18 [36] model with an initial learning rate of $10^{-3}$. Following [72], we keep the same learning rate for the first 100 epochs and linearly decay it to 0 over the remaining 100 epochs. We use Adam solver [73] with a batch size of 4, a $\beta_1$ of 0.5 and a $\beta_2$ of 0.99. We apply random flip and random rotation ($-15° \sim 15°$) for data augmentation.

**Comparisons with baselines.** For verifying the validity of the spatial–temporal module, we have designed one baseline method:

- Baseline: FCN-networks (BASE) and its improved variants with the spatial–temporal module:
- Proposed 1: FCN-networks + BAM (BAM);

- Proposed 2: FCN-networks + PAM (PAM).

All the comparisons use the same hyperparameter settings.

## 3. Results

In this section, we describe comprehensive evaluations on our proposed modules and comparisons with our methods and other state-of-the-art CD methods. Our experiments were performed on LEVIR-CD and SZTAKI datasets.

### 3.1. Comparisons on LEVIR-CD

We have compared BASE, BAM and PAM to verify the validity of the spatial–temporal module. All the comparisons use the same hyperparameter settings. Table 4 shows the ablation study of the baseline and its variants on LEVIR-CD test set. Precision, recall and F1-score related to the change type are computed for evaluating the performance of our method. We can observe that the spatial–temporal module (BAM and PAM) has a significant improvement over the baseline. Compared to the baseline, the BAM improves the F1-score by 1.8 points. Moreover, our multi-scale attention design (PAM) significantly improves the performance, with 1.6 points of F1-score improvement compared to the BAM.

**Table 4.** Ablation study of attention modules on LEVIR-CD test set.

| Method | Precision (%) | Recall (%) | F1-Score (%) |
|--------|---------------|------------|--------------|
| BASE   | 79.2          | 89.1       | 83.9         |
| BAM    | 81.5          | 90.4       | 85.7         |
| PAM    | 83.8 *        | 91.0 *     | 87.3 *       |

* PAM has achieved the best results.

Some change detection examples are displayed in Figure 6. There are many discontinuous small noise strips in the predictions (rows 2, 3 and 4 in Figure 6) of the baseline model. It is because the corresponding buildings in the two images can not be perfectly aligned, especially the edges of the buildings. Figure 7 better illustrates the misregistration errors. Our baseline model misdetects the misaligned parts of the building as the change area. We can observe that BAM and PAM models can well mitigate the misdetection caused by misregistration to produce smoother results. That is because when computing the response of the misaligned position, the spatial–temporal module learns the attention weights of the misaligned position and other positions. In this case, the response of the misaligned position gives less attention to the positions of the aligned buildings. Therefore, the misaligned position's response is less similar to that of the buildings. Besides, the baseline model fails to completely detect the building with a different color (row 5 in Figure 6) or a large scale (row 7 in Figure 6). The spatial–temporal module learns the global spatial–temporal relationships between pixels and exploits these dependencies to obtain a better representation. Therefore, BAM and PAM models are more robust to color and scale variations. Moreover, we can observe that PAM can obtain finer details than BAM and the baseline (rows 1 and 7 in Figure 6) due to its multi-scale design.

**Ablation study of batch-balanced contrastive loss.** Our batch-balanced contrastive loss (BCL) helps to alleviate the class imbalance problem. Table 5 shows the ablation study of BCL on LEVIR-CD test set. We can observe that our BCL gives consistency improvements to the performance of various models (BASE, BAM, PAM). When using BCL, the contribution of the minority (change) and the majority (no change) to loss is dynamically balanced in quantity during each training iteration, which reduces the possibility of the bias of the network towards a certain category and brings performance improvements.



**Table 5.** Ablation study of batch-balanced contrastive loss (BCL) on LEVIR-CD test set.

| Method | F1-Score (%) |
|---|---|
| BASE (without BCL) | 83.7 |
| BASE (with BCL) | 83.9 |
| BAM (without BCL) | 85.5 |
| BAM (with BCL) | 85.7 |
| PAM (without BCL) | 86.6 |
| PAM (with BCL) | 87.3 |

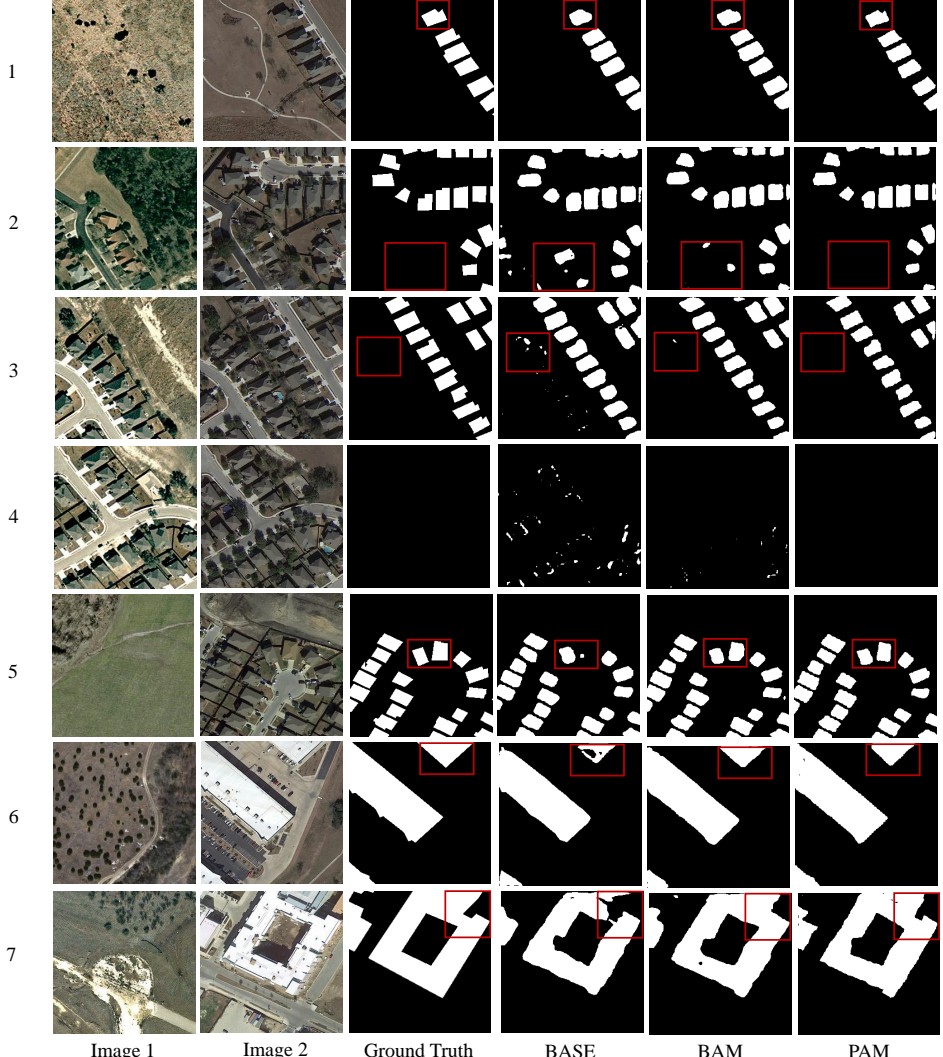

**Figure 6.** Change detection examples of our ablation experiments on the LEVIR-CD test set. The red boxes are drawn to highlight the advantages of our attention modules. Our BAM and PAM models obtain finer details (rows 1 and 7), with a lower false alarm rate (rows 2, 3 and 4), and higher recall (row 5, 6 and 7).

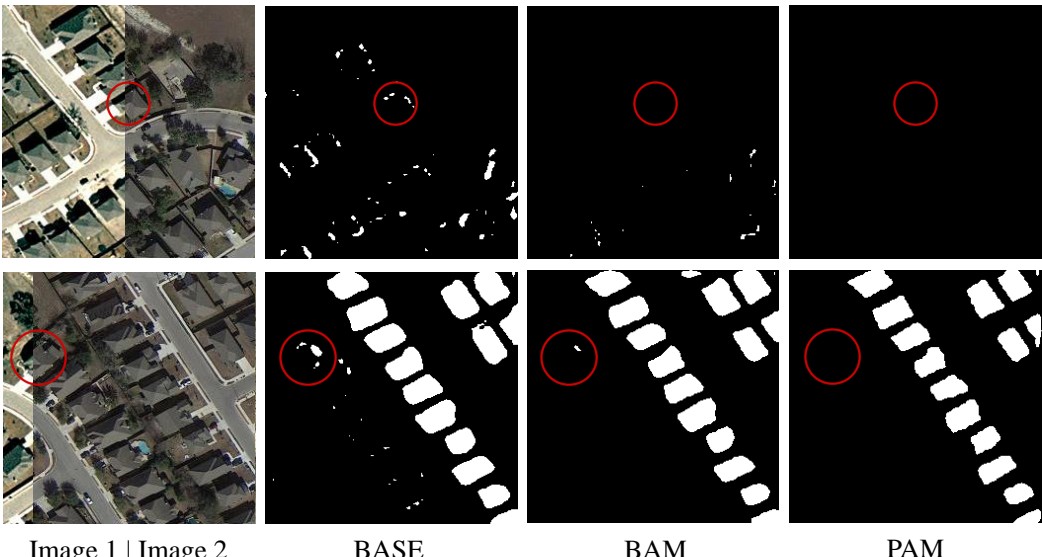

Image 1 | Image 2　　　　BASE　　　　　BAM　　　　　PAM

**Figure 7.** Visualization results of our ablation experiments on the samples suffering from misregistration errors. To visualize the misregistration errors, we stitch the two images. We use red circles to highlight the misregistration areas.

### 3.2. Comparisons on SZTAKI

We also evaluate the performance of our proposed method on the SZTAKI dataset and compare that with three state-of-the-art remote sensing image CD methods: TBSRL [28], rRL [74] and DSCNN [27]. In DSCNN [27], Zhan et al. designed a deep Siamese FCN model and used a weighted contrastive loss for CD in an end-to-end manner. The Siamese FCN consists of five convolutional layers without pooling and fully connected layers. In the testing phase, they utilized k-nearest neighbors to improve the initial change map generated by the deep Siamese FCN. In rRL [74], Huo et al. utilized the neighborhood relationship between training samples to enhance the overall separability of change features. However, the extracted image feature is handcrafted and lacks discriminative ability. In TBSRL [28], Zhang et al. employed Deeplabv2 [46] for extracting robust features and designed a triplet loss for learning the semantic relationship within the selected triplet examples. However, only the elements within the triplet are constrained by the semantic relationship, which lacks the exploration of global spatial information. Moreover, the spatial–temporal relationship is not well utilized either. We use the same training-testing split criterion as that used in [28]. Table 6 shows the comparisons of different methods on the SZTAKI dataset. Following [28], we report the performances on SZADA/1 (SZADA/1 denotes the first sample in the SZADA dataset) and TISZADOB/3 (TISZADOB/3 denotes the third sample in the TISZADOB dataset) separately for a fair comparison. The results of DSCNN, rRL and TBSRL are reported by [28]. We observe that our proposed methods (BASE, BAM and PAM) consistently outperform other state-of-the-art methods in F1-score. Figure 8 shows the change detection examples of different methods on the SZTAKI dataset. We can observe that our attention models can obtain more precise and smooth results than other methods.

**Table 6.** Comparisons of different methods on SZTAKI dataset.

| Methods | SZADA/1 | | | TISZADOB/3 | | |
|---|---|---|---|---|---|---|
| | Precision (%) | Recall (%) | F1-Score (%) | Precision (%) | Recall (%) | F1-Score (%) |
| DSCNN [27] | 41.2 | 57.4 | 47.9 | 88.3 | 85.1 | 86.7 |
| rRL [74] | 43.1 | 50.7 | 46.6 | 94.5 | 78.7 | 85.8 |
| TBSRL [28] | 44.4 | 61.9 | 51.7 | 86.0 | 93.8 * | 89.7 |
| BASE (ours) | 42.6 | 66.8 * | 52.0 | 94.3 | 86.4 | 90.2 |
| BAM (ours) | 44.6 | 64.2 | 52.7 | 91.8 | 91.4 | 91.6 |
| PAM (ours) | 45.5 * | 63.5 | 53.0 * | 95.0 * | 90.8 | 93.0 * |

* Best results.

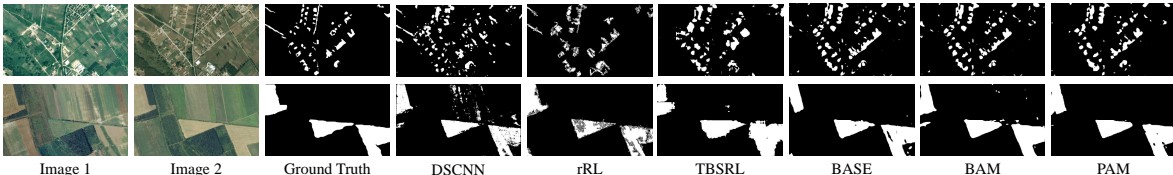

Image 1    Image 2    Ground Truth    DSCNN    rRL    TBSRL    BASE    BAM    PAM

**Figure 8.** Change detection examples of different methods on the SZTAKI dataset. Each row represents one sample and the predictions of different methods (first row: SZADA/1 data, second row: TISZADO/3 data).

### 3.3. Speed Performance

We tested our methods on a desktop PC equipped with an Intel i7-7700K CPU and an NVIDIA GTX 1080Ti graphic card. We used GPU to accelerate the training and testing process. Table 7 shows the time performances of different methods. We chose the state-of-the-art method, TBSRL [28] for comparison. However, it has not reported the time performance. As TBSRL adopts Deeplabv2 [46] with the ResNet101 [36] backbone as the feature extractor, we can infer the lower bound of its processing time by implementing the feature extractor module. In Table 7, the column of training time records time for training one epoch on the LEVIR-CD dataset. For a fair comparison, we adopt the same batch size (4) for all the experiments. Due to a well-designed network structure, our models take about 1 ∼ 4 mins for one epoch of training, which is consistently less than that of TBSRL. Our feature extractor utilizes the lightweight ResNet18 as the backbone, and concatenates pyramid-scale feature maps to fuse low-level edge information and high-level semantic information for efficiently generating dense feature maps, instead of using atrous spatial pyramid pooling (ASPP) [46] for producing dense feature maps. Our spatial–temporal attention module takes a long time for training because it needs to generate huge attention maps to measure the similarity between any two pixels, whose complexity of time is $\mathcal{O}((h \times w) \times (h \times w))$, where $h \times w$ is the size of the feature maps. PAM needs to calculate several attention maps of different pyramid levels, which takes more time than BAM.

**Table 7.** Time performances of different methods.

| Method | Training Time (s/epoch) | Testing Time (s) |
|---|---|---|
| BASE | 66 | 0.321 |
| BAM | 152 | 0.453 |
| PAM | 240 | 0.719 |
| TBSRL [28] | >342 | >0.9 |

Besides, Table 7 lists the testing time for an image pair of size 1024 × 1024 pixels. All of our models show better time performances than that of the lower bound of TBSRL. Notice that BAM only takes 30 percent more time than BASE, while PAM consumes about two times as long as BASE. We could design a PAM with a more concise pyramid structure (e.g., fewer pyramid levels) to balance the time

consumption and accuracy. Overall, our proposed modules have competitive time performances and acceptable time consumption.

## 4. Discussion

In this section, we interpret what the attention module learns by visualizing the attention module. Then we explore which pyramid level in PAM is the most important.

**Visualization of the attention module.** The self-attention mechanism can model long-range correlations between two positions in sequential data. We can think of bitemporal images as a collection of points in space-time. In other words, a point is in a certain spatial position and is from a certain time. Our attention module could capture the spatial–temporal dependency (attention weight) between any two points in the space-time. By exploiting these correlations, our attention module could obtain more discriminative features. For getting a better understanding of our spatial–temporal attention module, we visualize the learned attention maps, as shown in Figure 9. The visualization results in the figure were obtained by the BAM model, where each point is related to all the points in the whole space-time. In other words, we can draw an attention map of the size of $H \times W \times 2$ for each point, where $H, W$ are the height and width of the image feature map respectively. We selected four sets of bitemporal images for visualization; the first two rows in Figure 9 were from the LEVIR-CD test set; the other two were from the SZTAKI test set. For each sample, we chose two points (marked as red dots) and showed their corresponding attention maps (#1 and #2). Take the first row as an example: the bitemporal image shows the new villas built. Point 1 (marked on the bare land) highlights the pixels that belong to bare land, roads and trees (not building), while point 2 (marked on the building) has high attention weights to all the building pixels on the bitemporal images. As for the third row in Figure 9, point 1 is marked on the grassland and its corresponding attention map #1 highlights most areas that belong to grassland and trees in the bitemporal images; point 2 (marked on the building) pays high attention to pixels of building and artificial ground. The visualization results indicate that our attention module could capture semantic similarity and long-range spatial–temporal dependencies. It has to be pointed out that the learned semantic similarity is highly correlated with the dataset and the type of change. Our findings are consistent with that of non-local neural networks [35], wherein the dependencies of related objects in the video can be captured by the self-attention mechanism.

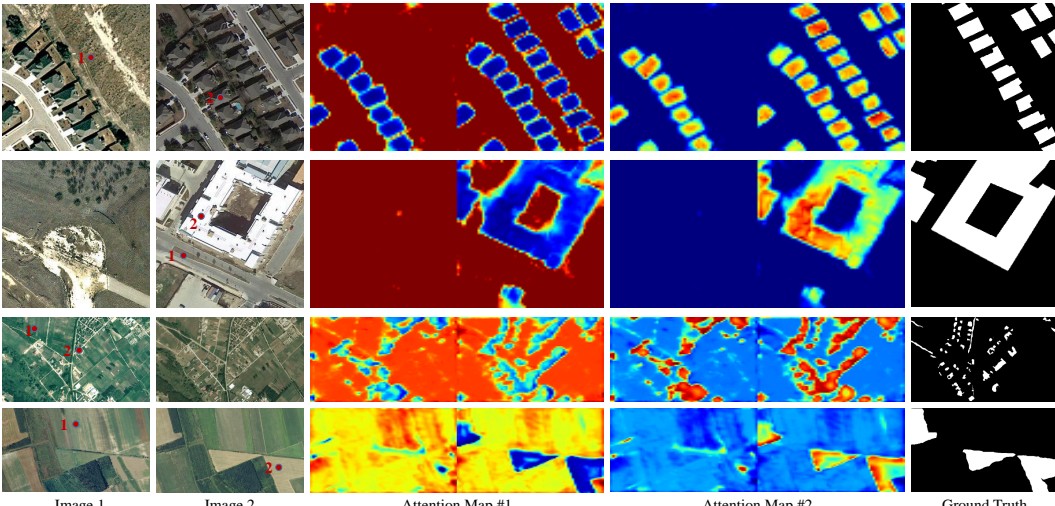

Image 1      Image 2      Attention Map #1      Attention Map #2      Ground Truth

**Figure 9.** Visualization of attention maps on LEVIR-CD and SZTAKI test sets. Each row represents one sample (image 1, image 2 and ground truth) and the visualized attention maps corresponding to the points marked on the input bitemporal images. For example, in attention map #1, it shows the attention of point #1 on all the pixels in the bitemporal images. Red denotes higher attention, while blue indicates lower attention.

**Which pyramid level in PAM is the most important?** The PAM combines attention features of different pyramid levels to produce the multi-scale attention features. In one certain pyramid level, the feature map is evenly partitioned into $s \times s$ subregions of a certain size, and each pixel in the subregion is related to all the bitemporal pixels in this subregion. The partition scale $s$ is an important hyperparameter in the PAM. We have designed several PAMs of different combinations of pyramid partition scales (1, 2, 4, 8, 16) to analyze which scale contributes most to the performance. Table 8 shows the comparison results on the LEVIR-CD test set. The top half of the table lists the performances of different PAMs; each contains only a certain pyramid level. We can observe that the best performance is achieved when the partition scale is 8. That is because in this case, the size of each subregion is agreed with the average size of the change instances. The input size of our network is $256 \times 256$ pixels, and the size of the feature map input to the attention module is $64 \times 64$ pixels. When the partition scale is 8, the size of each subregion in the feature map is $8 \times 8$ pixels, which corresponds to a region of $32 \times 32$ pixels of the input image for the network. Besides, the average size of the change instances in LEVIR-CD is 987 pixels, around $32 \times 32$ pixels. We also observe the poorer performance of the PAM with the partition scale of 16. That is because the attention region of each pixel becomes so small that it can not accommodate most change instances. The bottom half of the table shows the performances of PAMs with different combinations of several pyramid levels. The PAM (1, 2, 4, 8) produces the best performance by considering multi-scale spatial–temporal contextual information.

**Methodological analysis and future work.** In this work, we propose a novel attention-based Siamese FCN for remote sensing CD. Our method extends previous Siamese FCN-based methods [27,28] with the self-attention mechanism. To the best of our knowledge, we are the first to introduce in the CD task the self-attention mechanism, where any two pixels in space-time are correlated with each other. We find that misdetection caused by misregistration in bitemporal images can be well mitigated by exploiting the spatial–temporal dependencies. The spatial–temporal relationships have been discussed and proven effective in some recent studies [29,31], where RNN is used to model such relationships. We also find that PAM can obtain finer details than BAM, which is attributed to the multi-scale attention features PAM extracted. The multi-scale context information is important for identifying changes. A previous study [28] employed ASPP [46] to extract multi-scale features, which would benefit the change decision. Therefore, a future direction may be exploring a better way of capturing the spatial–temporal dependencies and multi-scale context information. We would like to design more forms of self-attention modules and explore the effects of cascaded modules. Additionally, we want to introduce reinforce learning (RL) into the CD task to design a better network structure.

**Table 8.** Comparisons between different combinations of several pyramid partition scales in PAM on the LEVIR-CD test set.

| Methods | Pyramid Scale | | | | | F1-Score (%) |
|---|---|---|---|---|---|---|
| | **1** | **2** | **4** | **8** | **16** | |
| PAM_1 | ✓ | | | | | 85.7 |
| PAM_2 | | ✓ | | | | 85.6 |
| PAM_4 | | | ✓ | | | 85.6 |
| PAM_8 | | | | ✓ | | 86.2 |
| PAM_16 | | | | | ✓ | 85.5 |
| PAM_1_2 | ✓ | ✓ | | | | 86.0 |
| PAM_1_2_4 | ✓ | ✓ | ✓ | | | 86.4 |
| PAM_1_2_4_8 | ✓ | ✓ | ✓ | ✓ | | 87.3 |

## 5. Conclusions

In this paper, we propose a spatial–temporal attention neural network for remote sensing image binary CD. We also provide a new dataset for remote sensing image CD which is two orders of

magnitude larger than existing datasets. The ablation experiments have confirmed the validity of our proposed spatial–temporal attention modules (BAM and PAM), which capture the long-range spatial–temporal dependencies for learning better representations. The experimental results show that misdetection caused by misregistration in bitemporal images can be well mitigated through our attention modules. Additionally, our attention modules are more robust to color and scale variations in bitemporal images. By extracting the multi-scale attention features, PAM can obtain finer details than BAM. We also visualize the attention map for a better understanding of our attention module. Our proposed methods outperform several other state-of-the-art remote sensing image CD methods on the SZTAKI dataset. Besides, our attention module can be plugged into any Siamese-FCN-based CD algorithms to introduce performance improvements. Finally, we hope that our newly introduced dataset LEVIR-CD will provide opportunities for researchers to develop novel, data-hungry algorithms for remote sensing image CD.

**Author Contributions:** Conceptualization, Z.S. and H.C.; methodology, H.C.; validation, H.C.; formal analysis, Z.S. and H.C.; writing—original draft preparation, H.C.; writing—review and editing, H.C. and Z.S.; funding acquisition, Z.S. All authors have read and agreed to the published version of the manuscript.

**Funding:** This work was supported by the National Key R&D Program of China under the grant 2017YFC1405605, the National Natural Science Foundation of China under the grant 61671037, the Beijing Natural Science Foundation under the grant 4192034 and Shanghai Association for Science and Technology under the grant SAST2018096.

**Conflicts of Interest:** The authors declare no conflict of interest.

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
