# Peer review of "A Spatial-Temporal Attention-Based Method and a New Dataset for Remote Sensing Image Change Detection"

_remotesensing, doi:10.3390/rs12101662_

Round 1
Reviewer 1 Report
I commend the authors on making a good revision. I think it can be accepted at this time.
Author Response
Many thanks for your very useful suggestion to improve the quality of our manuscript.
Reviewer 2 Report
Dear authors,
Your re-submission of your paper on “A Spatial-Temporal Attention-based Method and A New Dataset for Remote Sensing Image Change Detection” – has bee inmproving several aspects. The paper presents a new dataset (based on Google Earth images) to develop a change detection method. The topic is very relevant and not many labelled datasets exist that allow working on change detection. Maybe add more specifically, why did you select to use Google Earth (GE) images and also add the limitations of this choice. The change detection data set focus on buildings – which is very relevant as change detection methods often produce wrong change trajectories because of differences in imaging conditions.
The abstract has been improved – the only suggestion I would have is the final section “Our proposed attention module improves 3.4 points to the F1-score of our baseline model with acceptable computational overhead”. -> I would suggest to add the extact figure of F1 scores -> the F1 score improved from x to y.
The introduction is also much improved – I would add only the following related topics and references:
- There are also more recent studies on Markov random fields e.g., https://doi.org/10.1117/1.JRS.13.024514
- A study showed that FCNs can be trained to detect changes of a specific built-up class: https://www.mdpi.com/2072-4292/11/23/2844
- And the role of height information (of course this cannot be extract from GE image but could be added from other data): https://www.mdpi.com/2306-5729/4/3/105
Throughout the paper you often cut the text by figures – please put the figures after the end of a paragraph this makes the text much more easy to read -> for example Line 310 (but also at many other places).
I sill would advise shortening the following figure caption: Figure 1 and Figure 2 -> it is good to have figures self-explaining, but this text is a bit too long and fits better into the main text.
Materials and method: About the use of Google Earth images and the problems of different imaging conditions – you could also refer to this study which also faced issues and discussed them: https://www.mdpi.com/2072-4292/9/9/895/htm (but have the strong argument of using them in the context of resource constraint environments).
Discussion: part of this section is still presenting results, while the discussion of the results could be more specific. -> it would be good to relate your finding also to other studies.
Reviewer 3 Report
I really appreciate the effort of the authors in improving their manuscript. I have no further comments.
Author Response
Many thanks for your very useful suggestion to improve the quality of our manuscript.
This manuscript is a resubmission of an earlier submission. The following is a list of the peer review reports and author responses from that submission.
Round 1
Reviewer 1 Report
The authors propose a new change point detection algorithm, and also a new public database for change point detection. I commend the authors for releasing this data, which is a serious contribution to the remote sensing community.
However, I find the proposed method poorly motivated and hard to understand. Like most deep learning papers I read, the proposed method seems like a blackbox, with many choices poorly motivated. The performance seems good, but to me, this is not enough to warrant publication.
Despite this concern, I believe the paper may have merit, and the methodological content is not far below what appears regularly in Remote Sensing. I therefore recommend the paper be re-considered after the authors clarify their motivations and make their choices transparent.
Major Concerns:
1.) Basically, I am concerned that one can learn very little from the new method. It is really many methods combined, and as is common with much deep learning research, one wonders how much is really novel here, and how much is basically random perturbations of what is already know. To me, this is the biggest flaw of the paper. I recommend the authors radically alter their presentation, to make clear why what they are doing is so different and important, compared to all the other deep learning work that has flooded remote sensing. Without a good answer to this question, I cannot recommend acceptance.
2.) When discussing the new data set, many concerns appear to me. How was the new dataset captured? Over what times? Who annotated it? How were decisions about what to include made? As a member of the remote sensing community, I am grateful to the authors for releasing their data---this is an important but often thankless task. But, it is not really usable without being fully transparent with respect to all details of its generation.
3.) Equation (1) is totally unclear to me.
4.) Why remove should we remove the fully connected layers?
5.) Change point detection seems more natural as a soft assignment problem. Why did the authors consider hard assignment?
6.) The English is better than the typical Remote Sensing submission, but is still far from publication ready. I will comment on typos below, but I think a careful revision (or many, to be honest) would help the paper look really clean and sharp.
7.) Will the authors release their code, along with all scripts to reproduce all results?
Minor Concerns and Typos:
-“Handcraft” should be “handcrafted”
-“Supporting vector” should be “Support vector”
-“Contributions” should not be capitalized after “The” on line 86
-“Two orders” rather than “The two orders”
-“Magic” is probably the wrong word
-“Pixel wise” rather than “Pixel wisely”
-“Non-local”, not “None-local”
-Euclidean should be capitalized
Reviewer 2 Report
The manuscript presents a new method for change detection in remote sensing images. The study is interesting and quite thorough. I have only two minor suggestions: 1) English is generally understandable, but there are a few oddly constructed sentences and grammatical errors that could benefit from a careful revision by someone more knowledgeable on the English language. 2) Line 41: deep learning techniques indeed have been shown to be advantageous in many situations, but stating that they are always superior is not correct.
Reviewer 3 Report
This paper presents a deep learning approach for change detection (CD) in remote sensed images using an attention mechanism that is applied at multiple scales. A new dataset with images for CD is also introduced.
The proposed model makes use of existing methods but combines them in a novel way to capture spatio-temporal features at various scales to address CD. The approach is well presented, with detailed explanations and equations. Extensive references to related work and explanations of how the proposed approach differs from previous works are included. Experiments on the new dataset as well as another public dataset are presented to show that the proposed approach outperforms existing methods.
The discussion on the attention module should start out with a more intuitive description of the attention mechanism to give a more grounded explanation. Also, there are some misspellings and grammatical errors that should be fixed.
Overall, this is a well-thought-out paper that offers a sound approach and a significant dataset to the research area of CD in remote sensed images.
Reviewer 4 Report
Dear Authors,
Your paper on “A Spatial-Temporal Attention-based Method and A New Dataset for Remote Sensing Image Change Detection” – presents a new dataset (based on Google Earth images) to develop a change detection method. The topic is in principle very relevant and not many labelled datasets that allow to work on change detection exists. The change detection data set focus on buildings – which is very relevant as change detection methods often produce wrong change trajectories because of differences in imaging conditions. However, it is not very clear why you only focus on the change of buildings. This should be better motivated. The paper is interesting but rather difficult to follow and would require a good revision. I would suggest to following main improvements:
Abstract: Please revise the language of the abstract – it is not easy to understand and the abstract does not present specific results – also the overview of methods should be improved!
Introduction: Add some more background on the methods used in your paper and also add more literature e.g., line 72 etc the entire paragraph is without any reference while it refers to literature.
Figures: please shorten the figure captions and add the explanation/discussion of the figure to the main text.
Materials and method: the methods need to be better explained – it is not entirely clear how change is evaluated. For the new dataset LEVIR-CD, you need to add some more information. It uses Google Earth images, but are these images from the same sensor, which geographic location does the dataset covers. Furthermore, it would be important to get information on how the labelling was done. Why is only the change of buildings covered?
Results: The first part of the results is actually methodology?
Discussion: part of this section is still presenting now results, while the discussion of the results could be more specific.
Reviewer 5 Report
Review on “A Spatial-Temporal Attention-based Method and A New Dataset for Remote Sensing Image Change Detection”
In this paper, it is proposed a new CD procedure based on CNNs and a new dataset LEWIR-Cd to encourage the research on CD
General Comment.
I have to be honest in saying that this is a valuable work. However, I am wondering if this is the right journal for your submission. Please, consider the following general comments:
-
I went through an in-depth analysis of your bibliography and I was not able to find any reference related to the MDPI Remote Sensing Journal, even if this journal is in Quartile 1, according to the JCR 2018. Almost 64 % of your references are related to IEEE journal papers/conference proceedings/chapters. If you are not able to find references for your work in the MDPI Remote Sensing Journal, probably this is not the right journal for your work.
-
Change detection and publicly available dataset in your work. Although I think this is a very valuable work, Google earth images cannot be defined a proper remote sensing public dataset. Indeed they provide pre-processed images from satellites, aerial images etc. As far as I know, google earth images lack some features that catheterize a public remote sensing dataset (e.g. a clear channel definition, a clear image dynamic range, a clear spatial/temporal/radiometric resolution). A public available remote sensing database is, for example, the one related to the Copernicus program.
-
Introduction, methods and results are mixed to see the specific comments.
Specific comments:
-
Figure 1. Although this is useful to explain the concept. In my opinion results and introduction should be separated. Moreover, I would suggest to not anticipate your method in just a Figure. Indeed, it is not easy to understand what you have done at this stage of the manuscript.
-
Line 117, could you define the “C” in CxHxW?
-
Line 140-160 should be moved in the introduction
-
line 165. Could you define in the manuscript the meaning, in your work, of the query and the key vectors?
-
Section 2.1.2 not clear why the mentioned numbers have been chosen.
-
Line 180-181 “We feed X respectively into two different convolution layers (C , 1 × 1 / 1)”. Not clear what is the implication of this
-
lines 269 -298 should be part of the methods /data description section.